# Eco-Efficiency of the Fisheries Value Chains in the Gambia and Mali

**DOI:** 10.3390/foods10071620

**Published:** 2021-07-13

**Authors:** Angel Avadí, Ivonne Acosta-Alba

**Affiliations:** 1CIRAD, UPR Recyclage et risque, F-34398 Montpellier, France; 2CIRAD, Université Montpellier, Recyclage et risque, F-34398 Montpellier, France; 3CIRAD, UMR Innovation, F-34398 Montpellier, France; ivonne_alba@orange.fr; 4Evalivo, Sustainability Assessment of Food Systems, 02100 Saint Quentin, France

**Keywords:** environmental impacts, artisanal fisheries, fuel use intensity

## Abstract

The Gambian and Malian fisheries and fish processing value chains are predominantly artisanal and represent a key source of protein and livelihoods, yet their eco-efficiency has not been studied to date. A Life Cycle Assessment was used to estimate the associated environmental impacts of those value chains and provide information on the eco-efficiency indicators, which relate technical efficiencies to environmental impacts. The results showed that industrial Gambian fleets’ fuel use efficiency is rather low as compared with the global mean fuel use intensity (landed fish/consumed fuel) for both small pelagics and demersal fish. In Mali, the fuel use intensity of motorised artisanal fisheries is lower than the mean values for artisanal inland fisheries in developing countries, but the important increase of frozen imported fish from fish farming multiplies the estimated impacts by four. The least energy-intensive fisheries (cast nets and stow nets in Gambia and opportunistic fishers in Mali) feature better eco-efficiency scores. Based on the identified sources of inefficiencies, we suggest improvements in the landing/processing infrastructure and fishing units’ engines, coupled with technical and business training and improved processing methods, to ameliorate seafood eco-efficiency and a stronger recognition of the importance of the artisanal fisheries subsector to overcome challenges and improving resource management.

## 1. Introduction

Reducing the environmental impact of our food system is a requirement to preserve natural resources and our ability to produce food and respond to a growing demand [1]. In the context of declining fish stocks, growing population and increasingly complex relations among actors in the fisheries sector [2], a value chain analysis revealed strategies that enhance the sustainability and competitiveness of all economic agents involved [3]. Particularly in Africa, as large populations depend on fisheries for food security and livelihoods, studies combining socioeconomic and environmental ecosystem concerns are valuable to decision-makers [4,5].

In Africa, fish is also the main animal protein source (36%), and historically, this proportion has reached 40% for inland fish in the 1990s [6]. It has been recently estimated [7] that over 35 million coastal fishers live from this activity, with total landed catches worth 20 billion USD plus an additional 3.6 billion USD generated across value chains by the small-scale fishing sector alone. Nonetheless, the growth of fish supply in the region is generally lower than it should be to satisfy the growing demand [8].

The total fishery production in Africa was estimated at 9.4 Mt in 2010, 4.9 Mt of which were sourced from marine captures, 2.7 Mt from inland water fisheries and about 1.4 Mt from aquaculture [9]. The World Forum of Fishing Peoples estimated in 2017 that more than 60 million people rely on inland fisheries for at least part of their livelihood, and over 90% of inland waters are in Asia or Africa, while 55% of the catches are not identified by species [10]. The framework survey of inland fisheries conducted in 2012 by the West African Economic and Monetary Union showed that this subsector appears significant in terms of work, food self-sufficiency and economic value [11,12].

A multicriteria environmental assessment is the key to finding solutions and monitoring complex trade-offs and multiple ways for producers and consumers to reduce food’s environmental impacts [13]. A Life Cycle Assessment (LCA) evaluates the environmental impacts of a product or a service, including all the steps from the extraction of raw materials to the final user, disposal or recycling. Technical inefficiencies in food systems are often studied with LCA as an ISO-standardised framework [14] accounting for all resource consumption and emissions associated with a production system, between the economic system and nature in terms of environmental impacts, modelling environmental mechanisms to estimate the associated impacts. Those potential impacts are expressed through environmental indicators, such as t CO_2_ equivalents emitted, which contribute to climate change and, thus, to damages to human health and ecosystem quality.

Eco-efficiency, as defined by the World Business Council for Sustainable Development [15], is a useful concept to simultaneously address the economic performance of a productive activity and its relative level of impacts on the environment, including effects on exploited ecosystems in the case of extractive activities such as fisheries. The link between eco-efficiency and cleaner production, including in fisheries, has long been established [16,17]. The economic performance of fisheries and dependent value chains such as fish processing, fishmeal, etc. is driven by technical efficiencies.

Efficiency in fisheries is usually gauged in terms of the fuel use intensity (i.e., the ratio of fuel consumption to landed unit of fish) [18,19,20]. Other indicators include ratios between produced and invested values; some examples include energy and protein-to-energy [21,22]. More complex approaches include, for instance, the Data Envelopment Analysis, a linear programming methodology used to determine the relative (distance to target) efficiencies of a set of multiple comparable units [23].

Several LCA studies have explored the eco-efficiency of crops [24,25,26,27], livestock [28,29] and seafood systems [30,31,32] but few at the value chain level [33,34,35]. The goal of this work is to assess the environmental impacts and eco-efficiency of the Gambian and Malian fisheries-based value chains, to identify priorities for improvement and to offer recommendations aimed at improving the eco-efficiency and, thus, to reduce the environmental impacts of the concerned value chains while increasing their technical efficiency. An extensive mapping of the Gambian and Malian fisheries value chains was described by the authors [36,37,38]. As far as we know, the eco-efficiency of these fishery supply chains had not been explored before from such an angle, and, in general, LCA studies of African fisheries are seldom produced in contrast with other fisheries. The need for more publishing data on diverse seafood systems and countries has been highlighted in the literature [39].

This initiative intends to provide the evidence-based knowledge on the development impacts of value chain operations so as to assist decision-making for investment projects in agriculture/seafood and to facilitate sectorial policy dialogue. In such context, “value chain” refers to the network of economic agents or actors and their activities delivering a product or service, as understood in the context of the European Commission’s Value Chain Analysis (VCA4D) framework [4,5]. This understanding of the term is interchangeable with “supply chain”.

## 2. Materials and Methods

### 2.1. The Gambian and Malian Fisheries Value Chains

The Gambian fisheries value chain consists of a variety of fishing activities, dominated by “artisanal” (i.e., small-scale) production units and an important fish processing sector, which includes both artisanal and industrial transformation activities. The bulk of captures are destined for direct human consumption within the country, but a growing proportion is redirected to export-oriented processing, such as drying and smoking for direct human consumption in other countries and fishmeal production for indirect human consumption abroad. The most important species consist of small pelagics. Fishing vessels are highly specialised regarding fishing gears and targeted species.

Mali is a landlocked country described by the authors [40] as having one of the highest continental fishery productivities in the world. The Malian fisheries value chain is composed by actors that fulfil several activities (fishing, trading and processing) and by an important diversity of hydrological resources. The flood wave that annually runs through the Niger River changes the landscape, creating new fishing areas (e.g., floodplains, ponds, backwaters, etc.). Fishing activities are seasonally and geographically distributed with a great variability in catch the areas and species targeted. Fish is consumed either processed (smoked, dried, burned or oiled) or fresh. Many freshwater species are targeted, including catfishes and tilapias. Fishing in Mali is essentially artisanal, and in general, it is not specialised.

A complete inventory describing both value chains, their economic agents and products are needed as input for the environmental analysis. The primary data were collected during two fieldtrips in each country, carried out in 2019, and through a follow-up with key respondents.

### 2.2. Environmental Analysis Framework

LCA has widely been applied to study the environmental implications of seafood value chains [41]. The first step of LCA is to define the system boundaries and the goal and scope of the assessment. Then, an inventory of all input and output fluxes over the entire life cycle of the studied systems is established. The inventory was based on mainly relying on field-collected primary data (obtained via key informant interviews and structured questionnaires with both stakeholders and experts) and complemented with secondary data (scientific and grey literature). Based on the inventory, the calculation of relevant environmental indicators and interpretation of the results were the final steps.

#### 2.2.1. Goal and Scope

The goal of the LCAs was to estimate and compare the environmental impacts of a system or a product—in the present work, the Gambian and Malian fisheries-based value chains. Two main functional units to express impacts were retained: 1 t of fresh whole fish and 1 t of processed fish product (e.g., frozen, dried or smoked fish and fish portions). These functional units have been widely used in comparative seafood LCAs [39,42]. The reference flows are the annual landings per vessel category for fisheries and the amount of fresh fish required to produce 1 t of fish product for processing.

In LCA of fishery-based systems, including aquaculture and fish processing, special attention is given to inventory items, which are known to drive most impacts [39,41]. These items include energy consumption; fishing gear; feeds (aquaculture) and technical yields such as the fuel use intensity for fisheries, feed conversion ratio for aquaculture, fish-to-fishmeal yield for reduction industries and the fish-to-product yield for other fish processing.

Mass-weighted economic allocation was applied to assign fishery impacts between the landed target fish and by catch for fisheries featuring by catch. No impacts were allocated to fish residues from artisanal or industrial processing, as they are considered as wastes without a market price [43,44]. The technological scope encompasses the current technologies deployed in West Africa, and the temporal scope corresponds to the 2014–2018 period. The scope of the assessment was from cradle to market.

#### 2.2.2. Inventory of Value Chains

Representative units of production (UP), were constructed for each activity of the economic agents of the value chains (i.e., the various distinctive fisheries, artisanal and industrial processing and fish product distribution) for the most representative system types based on field data collection:Producers (artisanal and industrial fisheries, shellfish collection systems and fish farming for imported fish);Processors (artisanal drying, artisanal and industrial smoking, industrial freezing and fishmeal production);Distributors (in terms of fish transportation activities across trader types).

Ancillary systems, such as ice and fishing infrastructures (vessels and fishing gear), were also modelled. Background inventory data, including the provision of all fuels; grid electricity (Senegalese grid electricity retained as a proxy for the Gambian electricity provision); tap water and industrial inputs (chemicals, plastics and other materials) were obtained from ecoinvent v3.5 [45]. Scenarios of distribution were based on representative transport distances and means for both countries.

#### 2.2.3. Impact Assessment

The Impact assessment method ReCiPe (2.2 Endpoint World H/A (Hierarchy/Average)) was retained. Impacts were estimated for three areas of protection (AoP) based on many relevant impact categories [46] with different units (human health: disability-adjusted life years, ecosystems: species·yr and resources: USD). In ReCiPe, impacts on the AoP can be also expressed as dimensionless “points” (Pt) and combined into a single score after a normalisation and weighting step. These indicators only make sense in comparative contexts to express (a) the relative contribution of an impact category to the cumulative impacts of the product system on an AoP and (b) the cumulative environmental performance (impacts) of the product system.

The method Cumulative Energy Demand (CED) [47] was used to derive additional specific eco-efficiency indicators. This indicator includes renewable and not renewable energy used during the life cycle of products.

### 2.3. Eco-Efficiency Assessment

The VCA4D environmental analysis framework, based on LCA, was extended to assess the eco-efficiency of the studied value chains. Fuel use intensity (FUI), namely the ratio between landed fish and fuel consumed to catch and land the fish, is widely used as an indicator of fishery efficiency in a LCA, as it captures both the actual fishing effort as well as other components of the fishing activity, such as the fuel consumption associated with travel to and from fishing areas and even fuel-saving strategies and other skipper behaviours [39,48]. Fuel consumption and productive activities in general are determined in fisheries by multiple motivations of the fishers beyond simply profit maximisation [49].

Two additional indicators were used, energy return on investment (EROI) [21,22,30] and protein-per-impact (PPI) [22,35,50]. The former refers to the ratio of energy embedded in a fish product to the industrial energy (CED) required to produce it (expressed, for instance, with respect to its edible yield), while the latter represents the ratio of proteins (as a proxy for nutritional value) delivered by a product to the potential environmental impacts estimated by the LCA, as described in reference [22]. The relation between energy and protein obtained from fish, especially regarding inland fisheries, has been highlighted in the literature [51,52]. The gross energy content (GEC) was computed from the lipid and protein contents, following the energy contents of those components and equations presented in FAO documents [53,54]. The lipid and protein contents, and edible yields, were retrieved from several FAO sources [54,55,56,57].

## 3. Results

### 3.1. LCA of the Gambian and Malian Fisheries Value Chains

#### 3.1.1. Life Cycle Inventory Analysis

The system boundaries of the modelled systems are depicted in Figure 1. For the Gambian value chain, the main identified products were divided into two categories: on the one hand, fresh fish (small pelagic fish, demersal fish, shrimps and cephalopods from artisanal and industrial fleets) and, in the other, processed fish (dried fish, smoked fish, frozen fish, fish meal and braised-dried shellfish). In the Malian value chain, the products were less diversified, with fresh fish (for rural or urban markets from artisanal fishing) and processed fish (artisanal smoked fish for rural and urban markets and frozen, imported fish, including small pelagics or fish farming from Asia).

For each of these products, three types of actors were modelled—namely, producers (artisanal and industrial fisheries and shellfish collection systems); processors (artisanal drying, artisanal and industrial smoking and industrial freezing) and distributors (modelled in terms of fish transport activities). The inventories were modelled based on field surveys and the literature, as presented in Table 1.

For the Malian value chain, the actors modelled included: producers (Malian artisanal fishers who were categorised following the proportion of fishing activities in their total income as opportunistic, part-time, diversified and full-time fishers and foreign artisanal producers and industrial fishers based on the case of the Gambia and fish farmers of tilapia and clarias in China and Vietnam, respectively); processors (artisanal drying) and distributors (national and international transport to bring the products to market). The inventories were based on field surveys and were modelled as presented in Table 2.

#### 3.1.2. Environmental Impact Assessment

Results of environmental impacts are presented by the area of protection, followed by the contribution analysis per ton of product. Then, the single score (Pt) and Cumulative Energy demand (CED) in MJ are shown. Detailed impact assessment results, including midpoints indicators, are presented in the Appendix A.

##### Absolute and Relative Environmental Impacts

Impacts were calculated for three areas of protection: human health, ecosystems and resources (Figure 2). The impacts of both fisheries and processing activities, including the transport of fish products, predominantly affect AoP human health in both countries.

In the Gambia, impacts of the average fish captures from industrial fleets are higher than those of demersal artisanal fleets, disagreeing with the results from other studies [71], since, usually, industrial vessels benefit from economies of scale. Industrial fisheries, in particular, are very inefficient, with a large, normalised impact on human health. However, given the low percentage of industrial captures to the overall captures (17% vs. 83% by artisanal fleets), the impact of the average captured t of fish remains relatively low. The reasons of this specific dynamic in Gambian fisheries are multiple: the unusual shape of the Gambian Exclusive Economic Zone (EEZ) and the respective fishing zones for the artisanal and industrial fleets, the status of the targeted stocks limiting economies of scale and even the skipper effect limiting the performance of industrial vessels [48]. Moreover, the proxy Senegalese and Mauritanian data used to model industrial fisheries and fishmeal production may not be fully representative of the Gambian conditions.

The results of the fish-processed products in the Gambia (Figure 2b) showed that the fishmeal industry features lower impacts per t of product than all other fish processing activities, including artisanal ones. This is, in part, due to the other processing industries consuming mainly demersal fish, which features even higher fishery impacts than the small pelagics consumed by the fishmeal industry. Moreover, fish processing features important losses (cleaning residues, etc.), while the fishmeal industry does not.

In Mali, comparing all products (Figure 2c), imported fish and smoked fish in urban markets have the highest impacts. The uses of these products are rather different: smoked fish is consumed in small quantities as a spice, and its conservation lifetime is between 3 and 12 months. Frozen imported fish is sold in markets as a replacement of fresh fish, because of its competitive price and the packaging and transport conditions, which guarantee a much higher level of freshness than the current conditions in Mali local fish: outdated ice factories with a reduced supply in relation to demand coupled with inefficient storage conditions (old refrigerators and freezers out-of-order or otherwise non-operational). However, the potential environmental impacts are almost four times higher than local fish.

##### Contribution Analyses

The relative contribution of each element of the system to the estimated impacts is presented in a contribution analysis of the Gambian fish products (Figure 3). For most fishing activities, fuel consumption contributes to >90% of impacts (Figure 3a). The impacts of fish processing are determined mainly by the impacts of the supplying fisheries, except for shellfish processing, for which the main driver for the impacts is the combustion of wood as fuel (Figure 3b).

In Malian fisheries, as in the Gambia fisheries, fuel use is the main contributor to the estimated impacts when canoes are motorised (Figure 4a). Fuel is the source of 67–98% of the potential impacts, depending on the impact category. The motorisation rate is rapidly increasing. For the analysis, data from field surveys were used. The average amount of fuel used was estimated according to the proportion of motorised canoes per type of fishing unit.

The canoe fleets used by part-time and full-time fishers feature the motorisation of ~50% (one out of every two fishing units is motorised), while diversified fishers attain 10% and opportunistic fishers 0%, as they tend to use shore gear. Fuel consumption was multiplied by the motorisation rate, because the catch levels vary little with or without motorisation, as the limiting factor seems to be the equipment and the potential of the fishing areas. It was difficult to estimate a motorisation rate representative of all fishing areas, because according to the interviews, between 10% and 100% of canoes are motorised, whereas, according to the West African Economic and Monetary Union [12], only 10% of fishing units are motorised. The results by motorised/non-motorised canoes (Figure 4b) show the importance of this factor on the potential environmental impacts. Fishing gear is the following contributor; most nets and fishing tools are imported and made of nylon or other synthetic materials, including lead from recycled car batteries. In the last years, it has replaced traditional handmade gears, the materials of which were biodegradable. Even if most of materials are recycled at the end (fishing nets are woven into ropes), this point represents a point of attention.

The contribution analysis by type of product available on the Malian market (Figure 5) shows that the fishing activity is the main contributor to impacts, even for smoked fish, with processing having little impact in comparison. For imported frozen fish, the production of fish and the concentrate feed used are the most important in the case of fish from fish farms, followed by transport. For imported frozen small pelagic fish, it is the fishing and freezing stages that have the greatest impacts.

The transport of fish products in the Gambia has a relatively minor contribution to the overall impacts, even including the production of ice used in the transport of fresh fish (Appendix A), probably due to the relatively small road distances to be covered. The actual impacts are probable higher, due to the conditions of Gambian roads and vehicles, many of which probably do not fulfil the European emission standard “Euro 3” specifications, as is common in Africa [72,73].

In Mali, on the other hand, as distances are more important, the impact of fresh fish transportation is very considerable, representing up to 3.6 times the impact of landed fish (i.e., a single score of 64.7 Pt for fresh fish transported to urban markets vs. 18.0 Pt for the average t of landed fish, the latter dominated by landings by diversified fishers).

##### Single Score and Cumulative Energy Demand

A comparison of the environmental impacts expressed as a single score for all Gambian and Malian fresh and processed fish products are presented, respectively, in Table 3 with the estimation of the additional indicator Cumulative Energy Demand, which was used to calculate the energy return on investments and protein-per-impact, two eco-efficiency indicators.

In the Gambia, industrial fleets had the highest values for the CED indicator and for processed products was shellfish collecting and braising, while the potential estimated impacts by the LCA were the lowest. In Mali, the highest values for CED were obtained by frozen imported tilapia, where an important amount of energy was needed to produce a concentrated feed. These results confirmed that the environmental impacts of the whole value chain are driven by inefficient fuel (liquid fossil fuels, wood and other biomass) consumption.

### 3.2. Eco-Efficiency: Fuel Use Intensity, Energy Return on Investment and Protein-Per-Impact

#### 3.2.1. Fuel Use Intensity

Eco-efficiency indicators were computed for all landed fish and elaborated fish products from the two value chains. In the Gambia, the weighted average of artisanal pelagic FUI was 150 L/t and that of artisanal demersal was 401 L/t. The weighted average of industrial demersal FUI was 2055 L/t. The global mean FUI for small pelagic fish, in contrast, has been estimated at 42 L/t and that of demersal fish and cephalopods at 539 and 613 L/t, respectively [74]. If the contribution to total captures by industrial fleets is expected to increase, attention should thus be paid to improving their fuel use efficiency (e.g., reducing their FUI) and their potential contribution to stocks depletion.

For processed products in the Gambia, the environmental performance of fishmeal production is not driven by fuel consumption, as happens, for instance, for the Peruvian fishmeal industry [62], but by the FUI of the supplying fisheries. This can be explained by the important differences in FUI providing the fishmeal industry around the world: Peruvian purse seiners consume <20 L of fuel per t fish [58], while Gambian purse seiners consume 104 L/t and encircling gillnets 163 L/t. As a reference, mean African encircling gillnets feature a FUI of 31 L/t [75]. The fuel consumption of shrimp-targeting vessels, both industrial and artisanal, resembles those of the respective Senegalese fleets, as described in reference [61].

Fishmeal production, in particular, compares negatively with Peruvian and global fishmeal production, mainly due to the relatively higher impacts of raw material provision (Appendix A) and marginally to different technology levels. Peru, for instance, increasingly uses state-of-the-art indirect drying fuelled by natural gas [62] instead of the direct drying fuelled by heavy fuel most commonly used in West Africa (pers. comm. with various experts in the Mauritanian and West African fishmeal industry).

In Mali, the FUI of full-time fishers, diversified fishers and part-time fishers using motorised canoes reaches, respectively, 234 L/t, 70 L/t and 97 L/t. For processed products, the fuel consumption is the main driver of impacts, but it remains rather low compared to the FUI for artisanal inland fisheries in developing countries, which has been estimated at ~1200 L/t [76].

#### 3.2.2. Energy Return on Investment and Protein-Per-Impact

EROI and PPI are two eco-efficiency indicators that also give information about the food security of the value chains—in particular, for an important source of protein as fish. The composition data required to compute EROI and PPI is summarised in Table 4, and the computed indicators are presented in Figure 6. As expected, the less energy-intensive fisheries, as indicated by a lower FUI, feature higher (i.e., better) EROI. These fishing units include cast nets and stow nets in the Gambia and opportunistic fishers in Mali (Figure 6a). Moreover, as environmental impacts are generally correlated with FUI, higher (i.e., better) PPI are associated with the same types of fishing units. Regarding processed fish products, Gambian fishmeal production benefits from economies of scale, as reflected in both EROI and PPI (provided that the protein in fishmeal is not for direct human consumption and requires cycling through farmed fish). Smoked Malian products feature higher EROI and PPI values than Gambian ones, probably due to a combination of lower FUI of the providing fishing units and the use in Mali of cow dung to complement wood for fish smoking (Figure 6b). Gambian-processed shellfish features rather high PPI, due to relatively lower impacts associated with collection and braising. The EROI of fish products imported to Mali are low due to the energy expenditures of the refrigerated transport and energy needed to produce a concentrated feed. In contrast, for instance, Peruvian-cultured tilapia features EROI values of 4.3 (0.03 for frozen tilapia imported from China to Mali) [22].

### 3.3. Sensitivity and Variability

In the LCA and impact assessment in general, the identification of sources of uncertainty are an important step to validate the results. In artisanal fisheries, the FUI (which depends on both the average landings per vessel and its associated fuel consumption) is the main driver of uncertainty. In artisanal fish and shellfish processing, their processed volumes, yields, losses and fuel consumption are a key source of uncertainty, as actors seldom keep records, and thus, all data were obtained from recalls. Regarding industrial fisheries and fish processing, their historical and detailed data is not easily accessible, as the companies tend to be very secretive about it.

A large variability in fishery performances may be assumed, due to the seasonal and interannual dynamics of the targeted stocks, as well as the variations in fishing skills (including the so-called skipper effect [48], which influences the FUI and, thus, fuel use efficiency). These fisheries factors also affect the processing industries provided by fisheries.

## 4. Discussion: Priorities for Improvement

The value chains mapping presented in reference [36] already suggested several improvement directions, but the environmental analysis presented here allows to prioritise them from an environmental and eco-efficiency point of view. Overall, each function delivered by sub-chains in the two value chains features areas for improvement, particularly regarding fuel consumption for fishing and infrastructure for processing. Support and development programmes can help to reduce environmental and human health impacts. Liquid and solid fuels, in particular, are known to have a direct impact on human health via the release of particulates and other contaminants though combustion [77]. The assessments conducted excluded important sources of impact on oceans and other water bodies associated with fisheries, such as plastics and microplastics (from waste disposal at sea or gear loss, among other sources [78]) or bilge waters disposed of in the ocean, for which characterisation factors are generally lacking (until recently for marine litter [79]).

The improvements of landing and processing infrastructures, as well as the introduction of better fish processing methods, are clearly a priority from an eco-efficiency perspective. In both countries, inefficient fuel (liquid fossil fuels, wood and other biomass) consumption is the main driver of environmental impacts of the fishery-based value chains. As both countries exploit rather abundant stocks (more seasonally distributed in Mali), mainly by passive fishing gear, such efficiency gains in fishing units should come from improved/well-maintained engines and/or skipper training in fuel-saving strategies. For artisanal fish processing, improved infrastructure such as efficient smoking ovens [80] would reduce the consumption of biomass and reduce spoilage. Moreover, as widely discussed in the literature, technical and business training (e.g., record-keeping) is lacking among West African artisanal actors [81,82,83,84].

Globally, wild caught fish yields have stagnated, while the fishing effort has increased [85]. It has been reported that the CPUE of West African artisanal fisheries has been steadily decreasing (by one-third to date) since the 1950s and that artisanal CPUE is 11-fold less than industrial CPUE [86]. That means that (i) the fishing effort (i.e., time and fuel) has increased dramatically while catches have not; in other words, the efficiency has diminished, and (ii) that artisanal fisheries are one order of magnitude less efficient than industrial ones. This second statement is in contrast with the results presented. The low eco-efficiency of the industrial Gambian fishing units compared to other industrial fleets showed the need to assess not only the stocks and catch effort but, also, to include other environmental indicators from a LCA perspective, since industrial fishing has a spatial extent more than four times that of agriculture (>55% of the ocean area) [87].

Artisanal continental fisheries feature very few inputs, coupled with a high recycling rate of fishing materials and equipment. Despite a low level of investment, artisanal fishing in Mali contributes 7% to agricultural GDP and ensures food security and income for an increasing number of families. Nonetheless, the rapid trend towards the motorisation of canoes may increase environmental damage unless the same level of fuel intensity is maintained. The role of artisanal fishing must be recognised, and some authors call for prioritising this sub-sector [27] and its importance not only for subsistence but, also, in the lives of fishers in general [6] while recognising that the lack of management, regulation and monitoring are needed to avoid overexploitation.

Gambian fishing units are specialised according to the target species, and catches are mostly sold rather than consumed by the fishers’ families. In Mali, as in most of inland African countries, fishing is rather artisanal, and the catches are destined for local consumption. Fishers have developed different strategies with a wide range of simple and artisanal equipment to adapt to the variability of the Niger River changes. As in other regions, overexploitation, illegal and unreported activities, weak and uncoordinated institutions, lack of reforms and the untapped potential of small-scale fisheries are still challenges to overcome [7,9]. This sector features large gaps in the available data. For instance, fisheries and aquaculture statistics, especially those representing African small-scale activities, continue to lack in quality, as recently highlighted by FAO [88].

Fisheries management based exclusively in stock assessment and monitoring fishing effort disregards the behaviour of industrial fishing vessels that seems to be insensitive to the energy or seasonal costs but, rather, shaped by politics and culture [87] and, in the case of West Africa, the great influence of the environment over fish stocks, which are climate-driven [6,89] and sensitive to hydrological resources [90,91].

The import of frozen fish (wild caught small pelagics and cultured freshwater fish) is rapidly gaining importance in Mali and other landlocked African countries such as Zambia [92,93]. There is an ongoing trend and discussion about the imports of cheap Asian fish in Africa [93,94,95,96]. A recent study [37] showed the social, economic and environmental impacts of imported fish through the value chain.

To respond to the growing demand for fish, aquaculture is a potential pathway. For instance, FAO preconizes the further development of aquaculture in Africa as a priority [2]. The results presented here showed higher impacts of fish farming compared to local captures; thus, it seems that promoting intensive fish farming based on concentrated feed and intensive energy use (fuel and energy) will probably increase the environmental impacts of fisheries. For the sustainable development of aquaculture in Africa, a number of infrastructure, know-how and financial and technological challenges would need to be overcome [92,97,98,99].

## 5. Conclusions

The efficiency of Gambian and Malian fisheries, as well as the eco-efficiency of said fisheries and fish processing, were quantified by means of LCA-derived indicators. It was found that both metrics are relatively lower than the equivalent global processes and products, except for Malian inland fisheries. The main reasons for such performances include the type of fishery organisation and governance prevailing in these countries, which determines a continuous race for poorly regulated common-access resources. Moreover, migratory and/or high mobility fishing strategies increase fuel consumption. Improvements in the infrastructure and energy efficiency of fishing and processing units would likely contribute to improving the eco-efficiency of the fisheries-based value chain in both countries. Post-harvest losses, which directly impact food security, would be reduced with enhanced infrastructure and processing methods. The availability of data on these sectors is rather limited, and the activities are poorly monitored. Beyond the enhancement of the infrastructure, increased access to technical support, as well as the monitoring of indicators of pressure on resources, would allow for better knowledge and more effective support leading to a revitalisation of these value chains, coupled with improved environmental performances.

## Figures and Tables

**Figure 1 foods-10-01620-f001:**
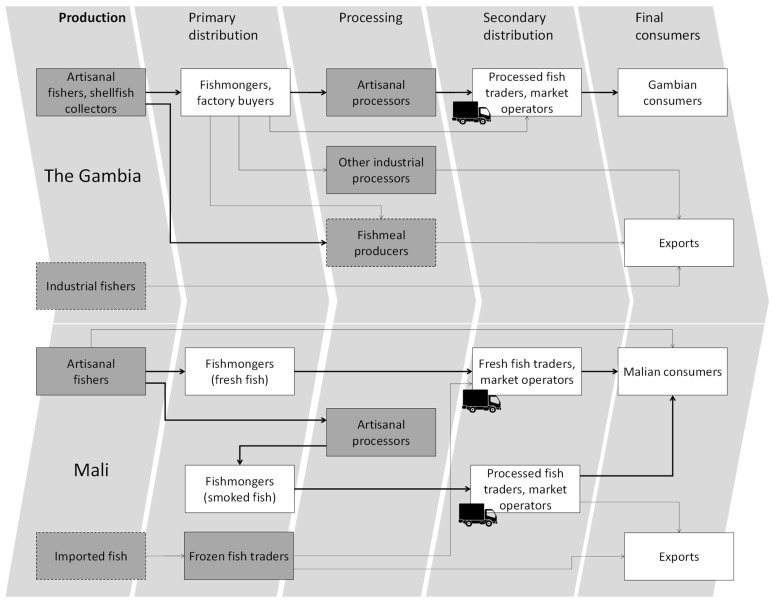
Simplified scheme of the Gambian and Malian fishery value chains (blocks in grey and transport icons represent systems modelled in the LCA; thicker lines represent more important flows; dotted boxes represent modelling based on secondary data and proxies.

**Figure 2 foods-10-01620-f002:**
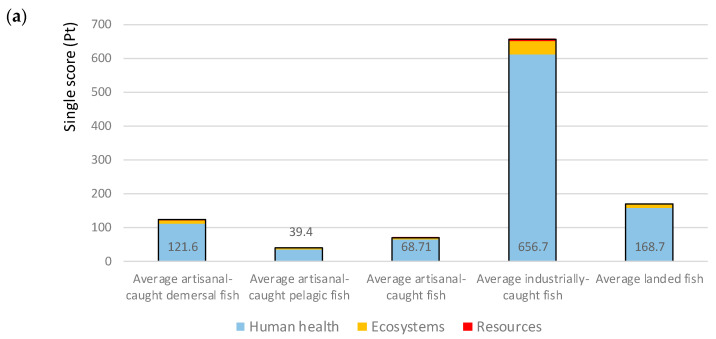
Endpoint environmental impacts per t of product and per area of protection: (**a**) average fish captures in the Gambia, (**b**) processed products in the Gambia and (**c**) average fish products in Mali.

**Figure 3 foods-10-01620-f003:**
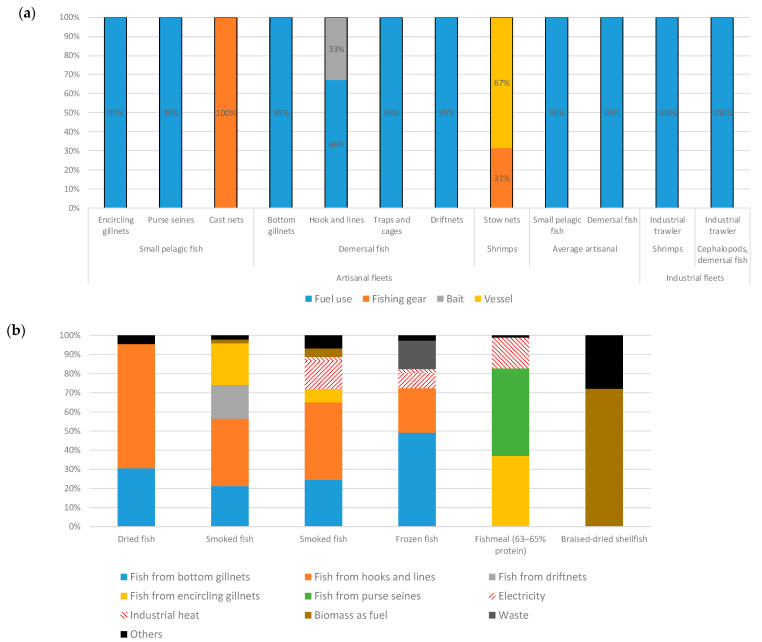
Contribution analysis of Gambian fish products per t of product and relative to the single scores of (**a**) fresh fish products and (**b**) processed fish products.

**Figure 4 foods-10-01620-f004:**
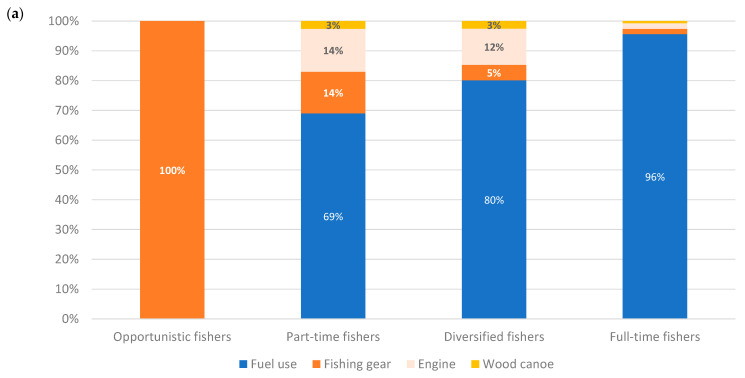
Contribution analysis of Malian fresh fish products per type of fisher, per t of product, relative to the single score, of those (**a**) non-segregated by motorisation and (**b**) segregated by motorisation.

**Figure 5 foods-10-01620-f005:**
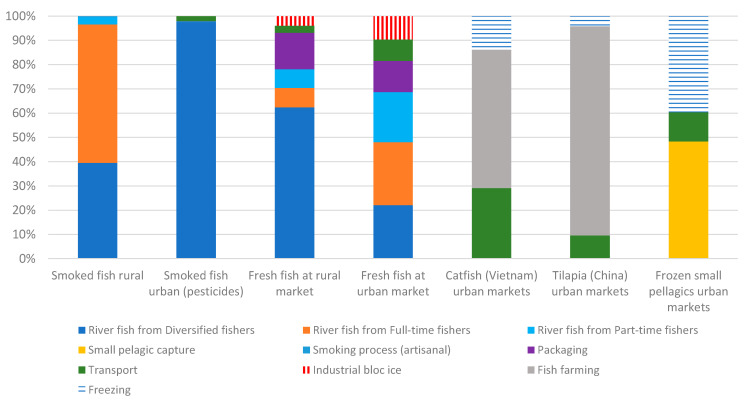
Contribution analysis of Malian transported fresh fish, processed fish products and imported fish products per t of product and relative to a single score.

**Figure 6 foods-10-01620-f006:**
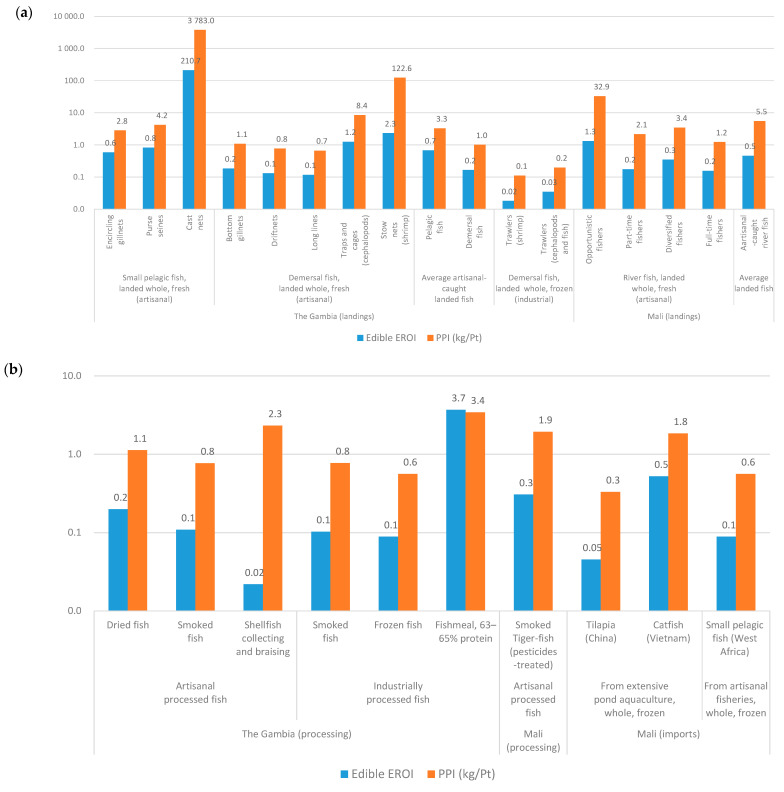
Edible energy return on the investment (EROI) and protein-per-impact (PPI) of Gambian and Malian fish products: (**a**) fresh fish and (**b**) processed fish.

**Table 1 foods-10-01620-t001:** Life cycle inventories by the type of unit of production in the Gambian fisheries value chain.

Unit of Production (UP) Type	Life Cycle Inventory Items
Fishing by fishing units	Hull and engine materials, fishing gear, fuel and lubricating oil consumption, ice, bait and refrigerant gases, antifouling paint and mean annual captures, landings and discards, as recommended in the literature [41,58,59]. Two-thirds of solids in the antifouling paint were assumed to be lost as antifouling emissions to the ocean, as usually assumed in fishery LCAs [59,60]. Secondary Senegalese data on industrial trawlers [61] was used, as most trawlers operating in the Gambia are Senegalese.
Shellfish collection	LCIs of shellfish collection are very basic, as no complex means of production are involved in that activity, except for dugout canoes, hand tools and fuel for braising. A 2-m-long dugout canoe weights up to 370 kg, and the amount of fuel consumed per UP (shellfish harvester) adds up to 1.7 m^3^ wood per year (sourced from Senegal and from commercial operations) for processing 273 kg shellfish flesh per year for a yield of 91 kg/year of dried shellfish products.
Production of ice	The production of ice for the artisanal fisheries is performed by both government-run and private plants. These plants produce between 1300 and 1600 t ice per year, consuming between 59.1 and 77.5 kWh/t ice.
Fish processing	Fish processing, both artisanal and industrial, were modelled in terms of fuels consumption (liquid fuels and biomass), chemicals and packaging materials, product yields and water and waste generation, as recommended in the literature [41,59].
Fishmeal production	Based on the secondary data obtained from West African (e.g., Mauritania) and global literature [41,62,63,64,65,66,67,68], as Gambia-based Chinese-owned fishmeal plants failed to provide data. Technical features and efficiencies of plants were considered at a level between Peruvian fair average quality and residual plants, with a fish-to-fishmeal conversion ratio of 4.5, and featuring direct drying fuelled by R500 residual fuel. The total amount of processed fish was computed based on the fish-to-fishmeal conversion ratio and export data, as all fishmeal is exported, and all exports are authorised by the Gambian Food Safety and Quality Authority (FSQA), which provided disaggregated data.
Fish distribution	In the Gambia, distances are relatively modest. As the bulk (67%) of artisanal landings destined for direct human consumption take place in the coastal landing sites, most of the fish is distributed through markets, and most of the population lives between the Atlantic coast and Farafenni, two transport distances that were estimated to represent the two main scenarios for fish products transport: the 25-km Tanji-to-Brikama segment and the 150-km Tanji-to-Farafenni segment. The impact of transporting fish upcountry was estimated based on these two reference distances; fish processing, both artisanal and industrial, was modelled in terms of fuel consumption (liquid fuels and biomass), chemicals and packaging materials, product yields and water and waste generation, as recommended in the literature [41,59].

**Table 2 foods-10-01620-t002:** Life cycle inventories by the type of unit of production of the Malian fisheries value chain.

Unit of Production (UP) Type	Life Cycle Inventory Items
Fishing by fishing units	Fishers were modelled as UP considering the canoe, equipment, fuel consumption, average catches, including seasonal variations, and losses. Small pelagic fishing was based on the processes developed for the Gambia. Farmed fish were modelled based on data from the literature, tilapia using inventories of Chinese production [69] and clarias produced in Vietnam based on the inventory in reference [70], considering the type of production (intensive or integrated fish farming), consumption of feed, electricity, diesel and lime per t of fish.
Production of ice	Modelled based on data collected in the field. The ice plants visited produced between 30 and 60 t of ice per day in 25-kg blocks, with an estimated average consumption of 50 kWh/t of ice.
Fish processing	Fish processing, both artisanal and industrial, was modelled in terms of fuel consumption (liquid fuels and biomass), chemicals and packaging materials, product yields and water and waste generation, as recommended in the literature [41,59].
Fish distribution	Transport distances in Mali are significant. For the fish that remained on the fishing grounds, only the use of the collection canoe was considered. To reach urban markets, fresh fish travelled an average of 540 km calculated by weighting the distances to the main fishing areas by the production volumes. Smoked fish travels 635 km (Mopti to Bamako). Frozen imported marine fish arrives by truck from the main ports (Mauritania and Senegal, average distance 1448 km) and from Morocco and Namibia (1371 km). Farmed fish is transported as refrigerated cargo by freighter from Asia (19,446 km and 16,800 km from China and Vietnam, respectively) to the port of Dakar and then by refrigerated truck to Bamako (1200 km).

**Table 3 foods-10-01620-t003:** Single score and cumulative energy demand (CED) of Gambian and Malian fish products per t of product.

Value Chain	Product	Production Units	Single Score (Pt)	CED (MJ)
The Gambia (landings)	Small pelagic fish, landed whole, fresh (artisanal)	Encircling gillnets	45.7	7437
		Purse seines	30.4	4946
		Cast nets	0.03	11.7
	Demersal fish, landed whole, fresh (artisanal)	Bottom gillnets	114.3	18,578
		Driftnets	190.8	34,871
		Long lines	190.0	31,190
		Traps and cages (cephalopods)	13.7	2407
		Stow nets (shrimp)	0.9	1289
	Average artisanal-caught landed fish	Pelagic fish	39.4	6415
		Demersal fish	121.6	20,353
	Demersal fish, landed whole, frozen (industrial)	Trawlers (shrimp)	1047.6	165,014
		Trawlers (cephalopods and fish)	632.3	99,596
Mali (landings)	River fish, landed whole, fresh (artisanal)	Opportunistic fishers	3.0	1568
		Part-time fishers	45.8	11,813
		Diversified fishers	28.6	5957
		Full-time fishers	80.1	13,300
	Average landed fish	Artisanal-caught river fish	18.0	4510
The Gambia (processing)	Artisanal processed fish	Dried fish	383.3	61,418
		Smoked fish	308.4	68,783
		Shellfish collecting and braising	77.4	216,061
	Industrially processed fish	Smoked fish	306.1	72,927
		Frozen fish	217.7	39,184
		Fishmeal, 63–65% protein	186.5	32,457
Mali (processing)	Artisanal processed fish	Smoked Tigerfish (pesticides-treated)	265.1	44,104
Mali (imports)	From extensive pond aquaculture, whole, frozen	Tilapia (China)	370.0	77,075
		Catfish (Vietnam)	73.4	16,651
	From artisanal fisheries, whole, frozen	Small pelagic fish (West Africa)	217.7	39,181

**Table 4 foods-10-01620-t004:** Fat, protein, gross energy content (GEC) and edible yield of the main species and fish products of the Gambia and Mali.

**Gambian Products**	**Lipid Content (%)**	**Protein Content (%)**	**GEC (MJ/kg) ***	**Edible Yield (%)**
Bonga shad	8.7	18.5	7.6	62
Flat sardinella	4.5	22.7	7.0	65
Round sardinella	4.5	21.0	6.6	59
West African ladyfish	5.3	23.0	7.4	57
Cunene horse mackerel	5.8	19.7	6.8	57
Common octopus	1.3	17.3	4.5	68
Sole fish/Tonguefish	1.4	19.8	5.1	68
Cuttlefish	1.4	18.0	4.7	63
Mullets	0.8	20.0	4.9	50
Pink shrimp	1.3	20.4	5.2	57
Dentex	2.5	19.9	5.5	52
Catfishes	1.9	21.3	5.6	52
White grouper	1.0	19.4	4.9	60
Sharks, skates, rays	4.5	21.0	6.6	70
Hairtails, Cutlassfishes nei	4.1	18.8	5.9	59
Dried fish	8.4	61.7	17.5	70
Smoked fish	7.4	33.9	10.7	70
Braised shellfish	1.5	18.0	4.7	100
Frozen fish	2.7	19.4	5.5	63
**Malian Products**	**Lipid Content (%)**	**Protein Content (%)**	**GEC (MJ/kg) ***	**Edible Yield (%)**
Bayad	1.2	17.1	4.4	60
African catfish	5.7	16.3	6.0	52
Freshwater catfish	5.7	16.3	6.0	52
African carp	0.6	15.7	3.8	54
Nile perch (Captain)	2.0	19.9	5.4	61
Tilapias	2.7	18.8	5.4	65
Mango tilapia	5.1	22.4	7.1	37
Smoked Tigerfish	6.2	73.3	19.3	70

* Gross energy content (GEC) was computed from the lipid and protein contents following the energy contents of said components and equations presented in the FAO documents [53,54]. The lipid and protein contents and edible yields were retrieved from several FAO sources [54,55,56,57].

## Data Availability

Raw primary data is not available due to confidentiality issues. All treated and anonymised data is available from the Appendix A.

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
