# Peer review of "Eco-Efficiency of the Fisheries Value Chains in the Gambia and Mali"

_foods, 2021, doi:10.3390/foods10071620_

Round 1
Reviewer 1 Report
This MS aims to use several indicators to evaluate the eco-efficiency of the fisheries value chains in The Gambia and Mali. The motivation and objectives are clearly stated and the MS is well structured which is worthy to be considered publishing in this journal. The authors used survey and literature research to conduct the analysis and results can provide useful information for future management reference. There are some questions need to be clarified. I suggest the authors take account the following comments and make a minor revision.
L 42, Please spell out Mt for the first time appearance.
L 69, CPUE is commonly used to present the density of fish stock rather than the efficiency in fisheries.
L 77, Lines 75-76 can be combined with the following paragraph. Why choose The Gambian and Malian fisheries as the target in this study should be mentioned.
L 83, The environmental impacts considered in the MS is mainly the fuel, however, marine debris from fishery such as discarded plastic bottles and lost fishing gears, etc also have impact on the environment. I suggest the authors mentioned this point in Discussion section at least.
Fig. 1, I don't understand why artisanal processors did not provide the products to fresh fish traders and why frozen fish traders imported fish and exported them for Mali? Please clarify.
L 229, Please define "MJ".
L 265, Yes, these products need more carbon foot print and have higher eco-impact.
L 272, Did the author consider the impact of catch, discard on the fish population or ecosystem? If not, this impact should be mentioned in Discussion section.
Discussion section
Different fishing types lead to different impact on fish resources. The impact on fish resources from different fishing activities need to be considered.
L 425, “less”.
L 465, But developing local aquaculture to solve the food security problem is worthy doing rather than importing from Asian countries which has higher eco-impact. The aquaculture technique of tilapia or catfish is pretty mature and easy. This approach can solve the problem of food security indeed.
Reviewer 2 Report
In this work, the authors assessed the impacts and eco-efficiency fisheries-based value chains. This manuscript contains the analysis of the assessment of the environmental impacts and eco-efficiency of the Gambian and Malian fisheries.
The authors included some suggestions on how to improve landing and processing infrastructure, and how to introduce better fish processing methods, and the necessity of technical and business training among artisanal actors.
This is the well-prepared report. I have only two comments.
The authors used inadequate citation patterns and inadequate style of references. Please see on Instruction for authors how it should be prepared.
Table 4. Is not the sense to repeat the unit % in the columns of Table 4. with every number.
